

# Order, please! Uncertainty in the ordinal-level classification of Chlorophyceae

Karolina Fučíková[1], Paul O. Lewis[2], Suman Neupane[2], Kenneth G. Karol[3] and Louise A. Lewis[2]

[1] Department of Natural Sciences, Assumption College, Worcester, MA, United States of America
[2] Department of Ecology and Evolutionary Biology, University of Connecticut, Storrs, CT, United States of America
[3] The Lewis B. and Dorothy Cullman Program for Molecular Systematics, New York Botanical Garden, Bronx, NY, United States of America

## ABSTRACT

**Background.** Chlorophyceae is one of three most species-rich green algal classes and also the only class in core Chlorophyta whose monophyly remains uncontested as gene and taxon sampling improves. However, some key relationships within Chlorophyceae are less clear-cut and warrant further investigation. The present study combined genome-scale chloroplast data and rich sampling in an attempt to resolve the ordinal classification in Chlorophyceae. The traditional division into Sphaeropleales and Volvocales (SV), and a clade containing Oedogoniales, Chaetopeltidales, and Chaetophorales (OCC) was of particular interest with the addition of deeply branching members of these groups, as well as the placement of several *incertae sedis* taxa.

**Methods.** We sequenced 18 chloroplast genomes across Chlorophyceae to compile a data set of 58 protein-coding genes of a total of 68 chlorophycean taxa. We analyzed the concatenated nucleotide and amino acid datasets in the Bayesian and Maximum Likelihood frameworks, supplemented by analyses to examine potential discordant signal among genes. We also examined gene presence and absence data across Chlorophyceae.

**Results.** Concatenated analyses yielded at least two well-supported phylogenies: nucleotide data supported the traditional classification with the inclusion of the enigmatic Treubarinia into Sphaeropleales *sensu lato*. However, amino acid data yielded equally strong support for Sphaeropleaceae as sister to Volvocales, with the rest of the taxa traditionally classified in Sphaeropleales in a separate clade, and Treubarinia as sister to all of the above. Single-gene and other supplementary analyses indicated that the data have low phylogenetic signal at these critical nodes. Major clades were supported by genomic structural features such as gene losses and trans-spliced intron insertions in the plastome.

**Discussion.** While the sequence and gene order data support the deep split between the SV and OCC lineages, multiple phylogenetic hypotheses are possible for Sphaeropleales *s.l.* Given this uncertainty as well as the higher-taxonomic disorder seen in other algal groups, dwelling on well-defined, strongly supported Linnaean orders is not currently practical in Chlorophyceae and a less formal clade system may be more useful in the foreseeable future. For example, we identify two strongly and unequivocally supported clades: Treubarinia and Scenedesminia, as well as other smaller groups that could serve a practical purpose as named clades. This system does not preclude future establishment

Corresponding author
Karolina Fučíková,
k.fucikova@assumption.edu

of new orders, or emendment of the current ordinal classification if new data support such conclusions.

## INTRODUCTION

Chlorophyta are a phylum of green plants comprising a variety of microscopic and macroscopic, uni- and multicellular, freshwater, terrestrial and marine algae that inhabit virtually every place on Earth that light and moisture can reach. The phylogenetic diversity of this phylum includes a number of deeply diverging lineages lumped under the term prasinophytes, and a group of core Chlorophyta, the majority of which falls into three classes (e.g., *Fučíková et al., 2014a*; *Fučíková et al., 2014b*). As genome-scale data became available for phylogenetic reconstruction, two of the classes, Trebouxiophyceae and Ulvophyceae, have been disputed in terms of their monophyly and internal classification (e.g., *Fučíková et al., 2014a*; *Fučíková et al., 2014b*; *Lemieux, Otis & Turmel, 2014*). Meanwhile, Chlorophyceae stands as the uncontested champion of monophyly, supported by molecular and ultrastructural data (*Mattox & Stewart, 1984*; *Lemieux et al., 2015*; *Fučíková, Lewis & Lewis, 2016*).

Within Chlorophyceae, two major sister clades are recognized –the SV and the OCC clade, composed of the orders Sphaeropleales and Volvocales (the latter is referred to as Chlamydomonadales in some sources), and Oedogoniales, Chaetophorales and Chaetopeltidales, respectively. These five orders and the phylogenetic divide between SV and OCC are well accepted and supported by molecular phylogenies and ultrastructural features (*Lewis et al., 1992*; *Turmel et al., 2008*; *Buchheim et al., 2012*; *Tippery et al., 2012*; but compare to e.g., *Pröschold et al., 2001*, which does not show the SV split but yields no support for the alternative topology). However, the phylogenetic distinctness of Sphaeropleales and Volvocales appears to fade when deeply diverging SV taxa are included in analyses (e.g., *Lemieux et al., 2015*, some analyses of *Tippery et al., 2012*; *Marin, 2012*). Among these *incertae sedis* taxa is the Treubarinia, which contains the genera *Treubaria*, *Trochiscia*, *Cylindrocapsa* and *Elakatothrix*, all morphologically divergent from each other and subtended by long branches in most analyses. Further uncertainly positioned taxa are the genera *Golenkinia* and *Jenufa* (*Němcová et al., 2011*). Examples of previously considered topologies are shown in Fig. 1.

Taxonomically, the most problematic is the placement of the family Sphaeropleaceae, a small group of genera that nomenclaturally defines Sphaeropleales. If Sphaeropleaceae were not to form a monophyletic group with the rest of taxa commonly treated as members of the order, Sphaeropleales would have to be split, likely to render Sphaeropleaceae as the sole family in the order. A new order would then be needed to accommodate the remaining families Bracteacoccaceae, Bracteamorphaceae, Chromochloridaceae, Dictyochloridaceae, Dictyococcaceae, Hydrodictyaceae, Mychonastaceae, Neochloridaceae,

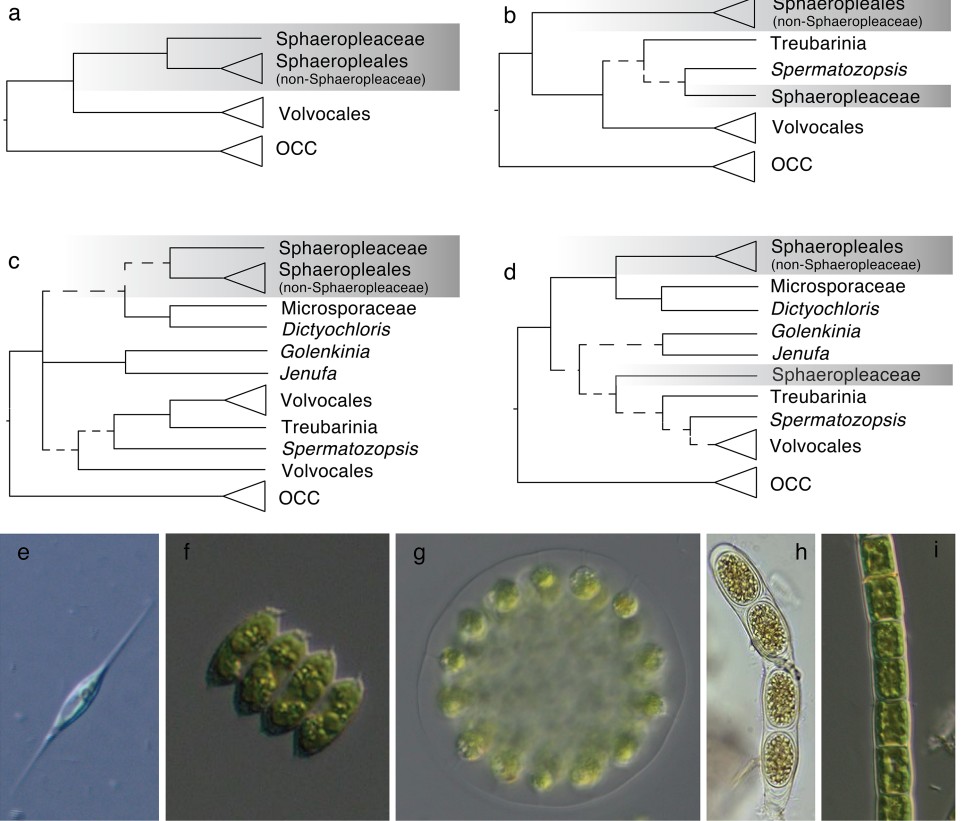

**Figure 1** **Phylogenetic hypotheses of ordinal classification within Chlorophyceae, based on previous and current analyses of ribosomal DNA.** (A) The traditional five-order scenario without *incertae sedis* lineages, based on *Buchheim et al. (2012)*, (B) a scenario showing non-monophyletic Sphaeropleales, based on *Marin (2012)*, (C) a scenario including additional incertae sedis based on *Němcová et al. (2011)*, and (D) a simplified tree based on our own 18S analysis (full tree shown in Fig. S3). Images on the bottom give examples of morphologies in the focal groups: (E) *Ankyra*, member of Sphaeropleaceae (F) *Desmodesmus* (non-Sphaeropleacean Sphaeropleales), (G) *Eudorina* (Volvocales), (H) *Cylindrocapsa* (Treubarinia), and (I) *Microspora* (*incertae sedis*). OCC is the clade containing the three orders Oedogoniales, Chaetophorales, and Chaetopeltidales. Dashed lines indicate low statistical support for clades. Triangles give a sense of species richness within groups—for example, Volvocales is a relatively large group whereas Treubarinia only contains a handful of genera and species. The shaded box highlights taxa currently included in Sphaeropleales.

Pseudomuriellaceae, Radiococcaceae, Rotundellaceae, Scenedesmaceae, Schizochlamy-daceae, Schroederiaceae, Selenastraceae, Tumidellaceae (*Fučíková, Lewis & Lewis, 2014a*) and Microsporaceae (*Tsarenko, 2005*), provided they all formed a clade (Fig. 1). Because Microsporaceae have not been firmly phylogenetically linked with Sphaeropleales (with the exception of unpublished 18S analysis by *Buchheim, Michalopulos & Buchheim (2001)*; no longer accessible online but cited e.g., by *Leliaert et al., 2012*), we here consider this family as another *incertae sedis* taxon.

It is often claimed that a more complete taxon sampling helps resolve phylogenetic problems by breaking up long branches (*Nabhan & Sarkar, 2012* and references within). Ribosomal DNA, and especially the 18S rDNA marker has been the most popular for

phylogenetic reconstruction for three decades, and thus offers the most complete taxon sampling currently possible (*Leliaert et al., 2012* and references within). The main drawback of 18S is its limited resolving power. For instance, the taxonomically well-sampled study of *Němcová et al. (2011)* showed monophyletic Volvocales (with the inclusion of the clade Treubarinia) and Sphaeropleales, but lacked statistical support for either of the two orders and many relationships within them. The case of the ribosomal internal transcribed spacer (ITS) is similar, even though some studies augmented the low number of characters in this marker by including secondary structure information into analyses (e.g., *Buchheim et al., 2012*). The longest nuclear rDNA gene, 28S, has also been shown as useful but likewise yielded low or inconsistent support for the key deep divergences in Chlorophyceae (e.g., *Buchheim, Michalopulos & Buchheim, 2001*). Nevertheless, rDNA-based topologies are good starting hypotheses for further phylogenetic examination (Fig. 1), and we here attempt to test them with new data.

In the last decade, chloroplast genome data have been used to recover robust phylogenies of green algae, leveraging the availability of 50+ genes evolving at a range of rates. The taxon sampling of chloroplast phylogenomic studies is also gradually improving, not only to include representatives of additional major lineages, but also strengthening the sampling within orders, families, and genera (e.g., *Turmel et al., 2008*; *Lemieux, Otis & Turmel, 2014*; *Lemieux et al., 2015*; *Fučíková, Lewis & Lewis, 2016*; *McManus et al., 2018*). Chloroplast genome-scale data thus appear particularly promising, as they balance a multi-gene approach with dense taxon representation, and especially as high-throughput sequencing becomes easier and more affordable, and bioinformatic tools for processing large data sets become more accessible.

Our study adds chloroplast genome-scale data from 18 newly sequenced taxa, covering the previously omitted genera *Uronema* in Chaetophorales, *Chaetopeltis* in Chaetopeltidales, and the species *Oedogonium angustistomum* in Oedogoniales (all OCC). We further present data from seven species in Volvocales, one from Sphaeropleales, and most importantly seven *incertae sedis* from the SV clade. Our analyses support a number of previously inferred phylogenetic relationships across Chlorophyceae, and fill several important sampling gaps in the SV clade. Based on our results we consider a broadened definition of the order Sphaeropleales to include Treubarinia and other *incertae sedis* taxa, and a competing phylogenetic hypothesis where Sphaeropleales are reduced to a single family, the Sphaeropleaceae. We discuss the conflicting results from different analyses and their implications for current and future taxonomic work at the ordinal level in Chlorophyceae.

## MATERIALS & METHODS

In the present study, chloroplast genomes of 18 taxa (Table 1) were obtained using Illumina HiSeq and MiSeq, yielding 100 bp and 250 bp paired-end reads, respectively. In addition to representatives of Volvocales and Sphaeropleales, we specifically targeted deeply diverging lineages in the Chlorophyceae, including *Cylindrocapsa geminella, Elakatothrix viridis, Trochiscia hystrix* (all are members of the *incertae sedis* clade Treubarinia), putative but uncertain Sphaeropleales affiliates *Microspora* sp., *Parallela transversalis* and *Dictyochloris*
**Table 1  Newly obtained partial and complete chloroplast genomes, their content summary, and their GenBank accession numbers.** Numbers for incomplete genomes represent recovered fragments only. Highly fragmentary genomes (recovered in > 20 contigs) are shaded in light gray; number of fragments indicated in parentheses under genome size. Asterisks indicate a completely sequenced genome. Introns in IR genes were only counted once. Most taxa were sequenced using HiSeq technology; MiSeq-sequenced genomes are marked with an M in the Taxon field.

| Taxon name | Culture collection and strain number | Order or clade | Genome size (bp) | % coding (intronic orfs excluded) | Number of introns | GC content (%) | GenBank accession number(s) |
|---|---|---|---|---|---|---|---|
| *Borodinellopsis texensis* | UTEX 1593 | Volvocales | 356,516 (8) | 27.5 | 11 | 33.2 | MG778120–MG778127 |
| *Chlorococcum tatrense** | UTEX 2227 | Volvocales | 242,172 | 42.3 | 17 | 36.1 | MG778173 |
| *Chloromonas rosae* | UTEX 1337 | Volvocales | 713,219 (11) | 14.6 | 22 | 33.7 | MG778174–MG778184 |
| *Chlorosarcinopsis eremi** | UTEX 1186 | Volvocales | 298,847 | 32.6 | 7 | 35.1 | MG778185 |
| *Desmotetra stigmatica* | UTEX 962 | Volvocales | 198,003 (5) | 40.1 | 2 | 30.6 | MG778230–MG778234 |
| *Palmellopsis texensis* | UTEX 1708 | Volvocales | 314,811 (45) | 27.2 | 5 | 42.0 | MG778446–MG778490 |
| *Protosiphon botryoides* | UTEX B 99 | Volvocales | 138,549 (9) | 55.7 | 7 | 30.1 | MG778491–MG778499 |
| *Follicularia botryoides* M | UTEX LB 951 | Sphaeropleales | 133,953 (59) | 42.0 | 3 | 33.8 | MG778351–MG778407 |
| *Dictyochloris fragrans* M | UTEX 127 | *incertae sedis* | 65,429 (66) | 50.4 | 6 | 30.0 | MG778235–MG778296 |
| *Microspora* sp. | UTEX LB 472 | *incertae sedis* | 212,651 (2) | 39.1 | 14 | 28.6 | MG778408–MG778409 |
| *Parallela transversalis** | UTEX LB 1252 | *incertae sedis* | 177,618 | 48.3 | 6 | 31.2 | MG786420 |
| *Spermatozopsis similis** | SAG B 1.85 | *incertae sedis* | 134,869 | 60.2 | 7 | 33.2 | MG778500 |
| *Cylindrocapsa geminella* | SAG 3.87 | Treubarinia | 107,144 (44) | 47.8 | 5 | 33.2 | MG778186–MG778229 |
| *Elakatothrix viridis* | SAG 9.94 | Treubarinia | 115,983 (55) | 43.9 | 10 | 27.8 | MG778297–MG778350 |
| *Trochiscia hystrix* | UTEX LB 606 | Treubarinia | 276,704 (32) | 32.2 | 19 | 32.0 | MG778501–MG778532 |
| *Chaetopeltis orbicularis* | UTEX LB 422 | Chaetopeltidales | 221,217 (48) | 33.8 | 20 | 27.7 | KT693210–KT693212 MG778128–MG778172 |
| *Oedogonium angustistomum* | UTEX 1557 | Oedogoniales | 147,210 (37) | 57.7 | 18 | 28.0 | MG778410–MG778445 |
| *Uronema* sp.* | CCAP 334/1 | Chaetophorales | 198,471 | 49.0 | 25 | 27.2 | MG778533 |

*fragrans* (identified as uncertainly positioned by *Fučíková, Lewis & Lewis, 2014a*), and the enigmatic putative volvocalean *Spermatozopsis similis*. Complete information regarding algal culturing conditions, DNA extraction, sequencing and annotation details can be found in *Fučíková, Lewis & Lewis (2016)*, and the most relevant details are included in our supplementary materials as well. GenBank accession numbers for all 68 taxa used in our phylogenetic analyses are shown in Table S1, along with taxonomic notes and information about gene content for each taxon's chloroplast genome.

## Bayesian analyses

Sequences of 58 protein-coding chloroplast genes were aligned using the translation-aided algorithm in Geneious v.10 (Biomatters) using the bacterial code and the remaining parameters set at default. Sites and regions of uncertain homology in variable genes were masked and trimmed prior to analyses using a custom Python script. We provide the masked alignments in supplementary files; the deploy.py script is available as supplement to *Fučíková, Lewis & Lewis (2016)*. Phylogenetic trees were constructed using MrBayes v.3.2 (*Ronquist et al., 2012*) using the GTR+I+ Γ model over 6,250,000 generations and two MCMC chains in each of two parallel runs. Sites were partitioned by codon position across the entire data set. The first 20% of each run was discarded as burn-in. The data
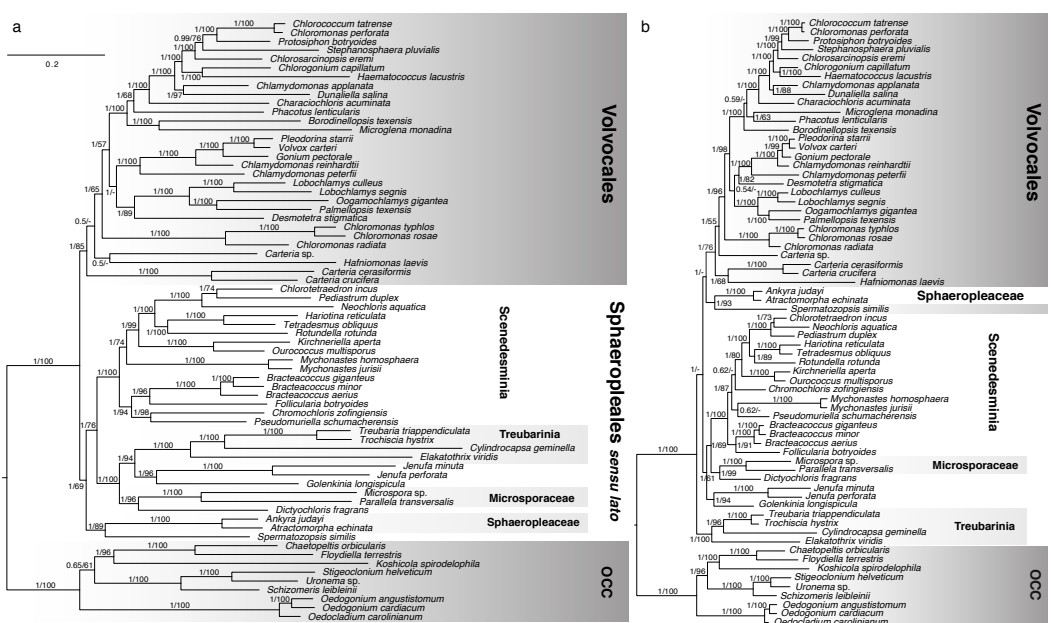

**Figure 2** Bayesian consensus trees inferred from analyses of concatenated (A) nucleotide and (B) amino acid chloroplast data (58 protein-coding genes). Taxon groups of interest are designated with boxes and tentative clade names. Scale bar represents expected number of substitutions/site for both trees. Numbers at nodes represent, respectively, Bayesian Posterior Probabilities (BPP) and Maximum Likelihood bootstrap values (BS) derived from 200 pseudoreplicates. BPP values lower than 0.5 and BS values lower than 50 are reported as dashes (-).

were analyzed in the amino acid (aa) form as well, implementing the aa GTR model in MrBayes.

Single-gene data sets were analyzed using MrBayes v.3.2 (*Ronquist et al., 2012*) as described above. A subset of these analyses (the 37 genes comprising the entire set of 68 taxa –i.e., with no missing data) was used to assess the degree to which individual genes support each clade in the phylogeny. One hundred randomly sampled post-burnin trees per gene were used for this assessment (3,700 trees total). The proportion of all 3,700 trees supporting each clade (PMT, or Proportion of Merged Trees), as well as the internode certainty (IC; *Salichos & Rokas, 2013*; *Salichos, Stamatakis & Rokas, 2014*), were calculated for the clades in Fig. 2A and plotted in Fig. 3.

## ML analyses

Maximum Likelihood (ML) analyses were carried out on the concatenated nt and aa data sets to complement the Bayesian analysis results. RAxML (*Stamatakis, 2014*) was chosen as analysis tool because of the size of our data set, but because RAxML does not allow the GTR amino acid model (which we consider the most realistic and therefore chose to implement it in MrBayes), we instead used the LGF model for the amino acid analysis, allowing for a gamma distribution of rates among sites (PROTGAMMALGF). The LGF model was selected using the script ProteinModelSelection.pl available from

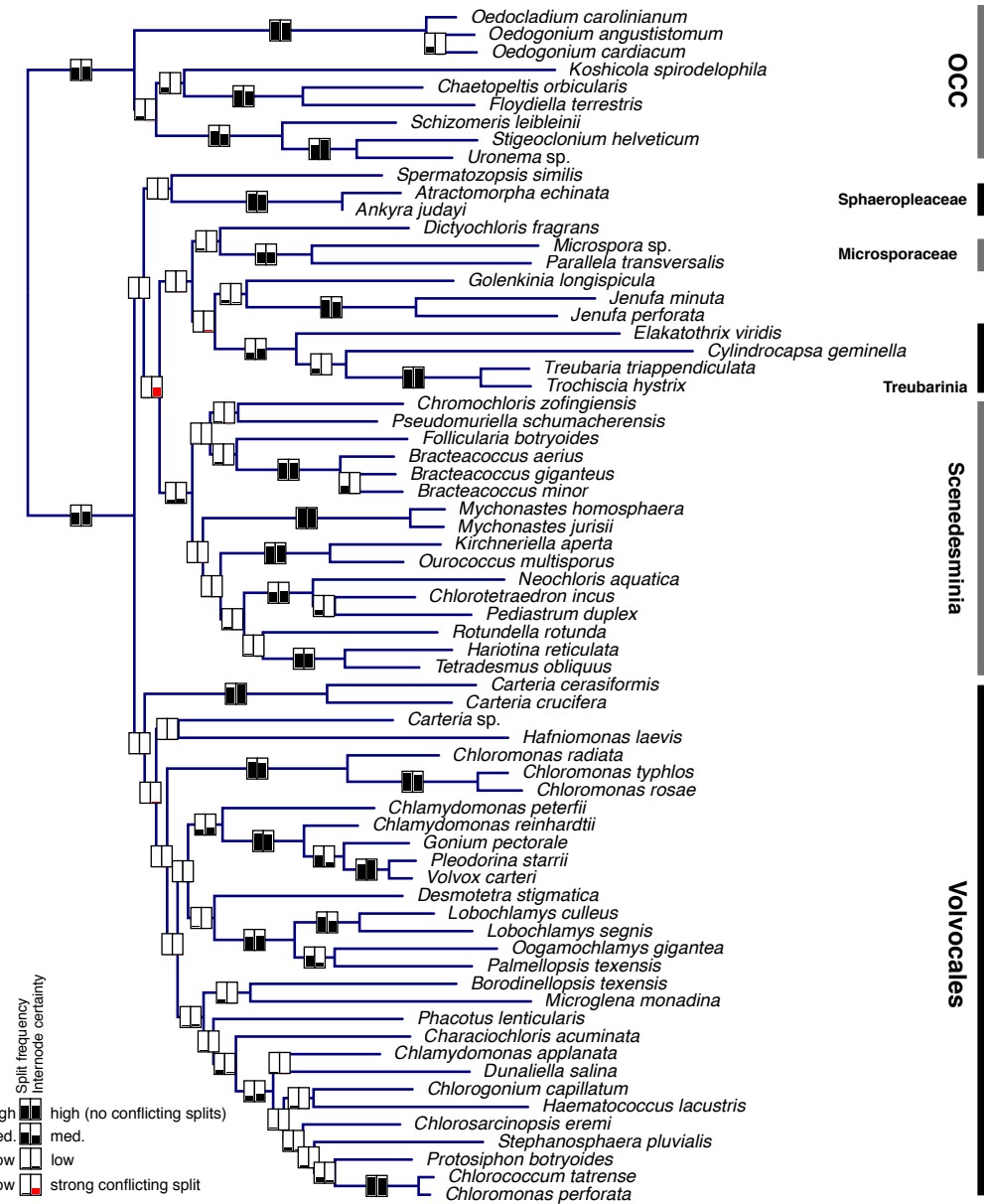

**Figure 3** Bayesian consensus tree from Figure 2a showing Proportion of Merged Trees (PMT) and internode certainty (IC) for each internal edge. Left box for each edge shows PMT and right box shows IC. For both PMT and IC, an empty box means 0 and completely filled box means 1. Red fill for IC means that the most important conflicting clade had a higher PMT than the one shown.

the RAxML webpage (https://cme.h-its.org/exelixis/web/software/raxml/hands_on.html). Two hundred bootstrap pseudoreplicates were carried out to assess branch support.

## Supplementary phylogenetic analyses

Because the initial concatenated analyses yielded conflicting topologies (nt vs. aa, Fig. 2), we used Shimodaira's Approximately Unbiased (AU) test (*Shimodaira, 2002*) within PAUP*

(*Swofford, 2002*) to see whether the two topologies were significantly different from each other. The test and its results are described in full in the Supplements.

On the concatenated data, a suite of analyses was conducted to probe the effects of analysis type and model selection on the final topology. To allow site-specific evolutionary processes in tree inference, we analyzed the nt data using SVDquartets (*Chifman & Kubatko, 2014*; *Chifman & Kubatko, 2015*) with 200 bootstrap pseudoreplicates implemented in PAUP* (*Swofford, 2002*).

We also carried out a supplementary set of Bayesian analyses with ambiguously recoded Serine, Isoleucine and Arginine codons (as described in *Fučíková, Lewis & Lewis, 2016*) to probe the possible effects of convergent codon usage. Further, we analyzed the 1st and 2nd positions only, and in a separate analysis the 3rd positions, to determine whether there was conflict between these data partitions. The data sets with analysis specifications and the resulting consensus trees are in the Supplements. The full concatenated data set was also analyzed using PhyloBayes v.4.1 and implementing the CAT-GTR model, allowing (in addition to gamma-distributed rates) site-specific substitution processes and thereby attempting to further mitigate branch attraction and other systematic bias issues in the data (*Lartillot & Philippe, 2004*; *Lartillot, Lepage & Blanquart, 2009*). The commonly used 18S nuclear ribosomal gene was analyzed to provide a chloroplast-independent estimate of the phylogeny. Because of the unique alignment issues associated with this gene, we selected the program BAli-Phy v.3, which estimates the optimal alignment as well as the phylogeny (*Suchard & Redelings, 2006*). The GTR+I+ Γ model was implemented and 12,000 iterations were run after pre-burnin. Parameter stability was checked using Tracer v1.7 (*Rambaut et al., 2018*) and 20% of the run were discarded as burnin. Lastly, an analysis combining the 18S and plastid nucleotide data was conducted, and is described in full in the supplementary methods.

We used TREESPACE (*Jombart et al., 2017*) to visualize the variability of posterior trees among genes in 2-to 3-dimensional Euclidean space. In this approach, the pairwise distances between posterior trees were computed (as Robinson-Foulds unweighted metric, *Robinson & Foulds, 1981*) from the package phangorn, and decomposed into a low-dimensional Euclidean space using metric multidimensional scaling (MDS). Before applying MDS, TREESPACE transformed the unweighted Robinson-Foulds tree distances into Euclidean distances using Cailliez's transformation (*Cailliez, 1983*). The analysis was performed on posterior trees from 37 genes comprising an entire set of Chlorophycean taxa (68 spp.) sampled in the study, corresponding to the data used for Fig. 3. TREESPACE was performed on the collections of 3,700 posterior trees (representing 100 randomly sampled trees/gene) obtained from MrBayes MCMC analysis.

## RESULTS

An overview of gene content in the context of other chlorophycean chloroplast genomes is presented in Table S1. For most of the 18 newly sequenced taxa, the full cp genome was not possible to assemble from the data, but full sequences of all or nearly all protein-coding genes were recovered nevertheless, as were most of the rRNA and some of the tRNA genes.

**Table 2   Summary of phylogenetic support for taxonomic groups of interest derived from concatenated chloroplast (cp) analyses and from the 18S BaliPhy analysis.** All Bayesian posterior probability values were converted to percentages; RAxML values are derived from 200 bootstrap pseudoreplicates. Blank cells indicate that the grouping of taxa was not present in the consensus tree.

| Analysis software | Data | Sphaeropleales s.l. | Scenedesminia | Volvocales + Sphaeropleaceae + *Spermatozopsis* | Sphaeropleaceae + *Spermatozopsis* | *incertae sedis* clade | *Jenufa* + *Golenkinia* |
|---|---|---|---|---|---|---|---|
| MrBayes | cp nt | 100 | 100 | | 100 | 100 | 100 |
| | cp aa | | 100 | 100 | 100 | | 100 |
| | cp nt 1st and 2nd positions | | 100 | 70 | 100 | 100 | 100 |
| | cp nt 3rd positions | 50 | | | | 100 | |
| | cp nt ambig | 100 | 100 | | 100 | 100 | |
| | cp nt + 18S | | 100 | 97 | 100 | 100 | 100 |
| RAxML | cp nt | 69 | 100 | | 89 | 100 | 96 |
| | cp aa | 54 | 100 | | 93 | | 94 |
| PhyloBayes | cp nt | | 100 | 100 | | 100 | 100 |
| | cp aa | | 100 | 73 | 99 | | 100 |
| SVDquartets | cp nt | | 100 | | | 97 | 72 |
| BaliPhy | 18S nt | | 84 | | | | 57 |

The concatenated nucleotide (nt) data set comprised 34,422 nucleotide and gap characters and 68 taxa after trimming. The amino acid (aa) data set had 11,474 characters. Given the multitude of analyses conducted here, the main results are summarized in Table 2, which shows the changing support for main groups of taxa across the different analyses.

The four-genus clade Treubarinia was recovered as strongly monophyletic in all concatenated chloroplast analyses. In most of our analyses, the genera *Golenkinia* and *Jenufa* also consistently grouped together, albeit with varying support. In the Bayesian framework, both analysis types (nt and aa) yielded a tree with a monophyletic OCC clade and the majority of Volvocales supported as monophyletic (Fig. 2). *Spermatozopsis* (traditionally considered a member of Volvocales) received high support as sister to the family Sphaeropleaceae, represented by *Atractomorpha* and *Ankyra*, and the coccoid *Dictyochloris* strongly grouped with Microsporaceae, represented by *Parallela* and *Microspora*. The nt tree shows monophyletic Volvocales, in which *Carteria* strains and *Hafniomonas* are the deepest diverging lineages. The OCC clade and the three orders within it were recovered consistently with previous studies, with the addition of *Uronema* as sister to *Stigeoclonium*, *Chaetopeltis* as sister to the previously available *Floydiella*, and *Oedogonium angustistomum* as sister to *O. cardiacum* as expected. Sphaeropleales formed a strongly supported clade with *incertae sedis* taxa, featuring Sphaeropleaceae + *Spermatozopsis* as the deepest diverging group, and Treubarinia plus the remaining *incertae sedis* forming the sister group to the rest of "traditional" Sphaeropleales (Scenedesmaceae, Selenastraceae, Bracteacoccaceae, etc.), which we hereafter call Scenedesminia. The result of the ambiguated analysis (Fig. S1) was largely consistent with the nt tree.

Minor differences between the nt and aa phylogeny (Fig. 2) include the relative position of *Phacotus*, *Microglena* and *Borodinellopsis*, the positions of *Desmotetra* and *Carteria* sp. –all within Volvocales. Further, within Sphaeropleales the relative positions of *Chlorotetraedron*, *Neochloris*, and *Pediastrum* differ between the two analyses, as well as the placement of the coccoid genera *Chromochloris*, *Pseudomuriella* and *Mychonastes*. Major topological differences between the two trees involve the monophyly of Sphaeropleales, in particular, the position of the type family Sphaeropleaceae relative to the rest of taxa commonly placed in the order (Scenedesminia). The aa analysis shows Sphaeropleaceae grouping with Volvocales with absolute Bayesian support. Contrastingly, the nt tree has Sphaeropleaceae at the base of the clade containing the rest of Sphaeropleales and *incertae sedis* taxa including Treubarinia. We will refer to this clade as "Sphaeropleales *sensu lato*" or "Sphaeropleales *s. l.*" from now on (Fig. 2). Another major difference is the position of Treubarinia in the aa tree, where the clade is placed as sister to all non-OCC chlorophyceans.

The concatenated ML analysis yielded a consistent topology and comparable amounts of nodal support to the Bayesian tree in case of the nt data. The aa data set analyses, however, yielded different results in the ML framework when compared to the Bayesian results. The ML aa analysis (analyzed under a different model, dictated by the available options in RAxML; full results in supplements), was more consistent with the nt analyses, in that it showed monophyletic Sphaeropleales *s. l.*, albeit with weak bootstrap support of 54. Within Sphaeropleales *s. l.* the best ML aa tree (supplementary files) had a different topology from any other concatenated analysis conducted in our study. The Scenedesminia were recovered as monophyletic, but with Microsporaceae plus *Dictyochloris* as sister clade, and Sphaeropleaceae + *Spermatozopsis* as the next closest clade (this relationship was however not supported by the bootstrap analysis), *Jenufa* + *Golenkinia* as next still, and Treubarinia as the deepest diverging lineage in Sphaeropleales *s. l.*

Individual gene analyses of nucleotide data provided consistently strong support for many clades (Fig. 3, Supplementary Information) but showed a striking lack of support for the deepest clades, with the exception of the OCC clade. Internode certainty (IC) values were generally positive and correlated with the proportion of merged trees (PMT) support values, with the exception of the Sphaeropleales clade (including *incertae sedis* taxa but excluding *Spermatozopsis* and Sphaeropleaceae), which had substantial negative internode certainty (−0.535) and low PMT (0.007). The conflicting clade causing this negative IC places *Spermatozopsis* with the two *Mychonastes* taxa (PMT 0.062).

Fig. 3 clearly shows that most of the consistent phylogenetic signal in the nucleotide data is concentrated in relatively shallow parts of the tree. It is important to point out that the analysis depicted in Fig. 3 does not preclude the possibility that individual genes strongly conflict with one another with respect to the deep nodes. If the concatenated tree represents the least objectionable tree topology (i.e., a topology that does not include any group strongly contested by at least one gene; see discussion of Figure 5 and Table 5 in *Lewis et al., 2016*), then one would expect contentious clades to have low support. It seems safe to conclude that we can be confident in concatenated tree clades that also have strong PMT and IC because these are supported by individual genes as well as the concatenated data set. Examination of single gene trees using Treespace (Supplementary Information) contributes

further evidence—in some cases, such as in the case of Sphaeropleales *s. l.* (a low-confidence node in Fig. 3), the concatenated trees occupy a part of the two-dimensional treespace distant from the single-gene analyses (visual example shown in Fig. S6. A similar treespace disparity (concatenated vs. single-gene) is found for Volvocales + Sphaeropleaceae + *Spermatozopsis*—a grouping that receives support in amino acid analyses and in both PhyloBayes analyses (Fig. 2, Table 2, Supplementary materials), and interestingly also for Scenedesminia, which, however, does not exhibit among-gene conflict (Fig. 3).

The SVDquartets tree (Fig. S2) shows a monophyletic OCC clade, but with *Koshicola* as sister to Chaetophorales and Chaetopeltidales, rather than diverging at the base of Chaetopeltidales. The tree also shows monophyletic Volvocales (incl. *Hafniomonas* and all *Carteria* strains) albeit with low bootstrap support of only 41. Sphaeropleales receive similarly low support of 49 and include *Spermatozopsis* not as sister to Sphaeropleaceae (as was the case in MrBayes analyses) but instead as sister to Scenedesminia (BS support 58). A strongly supported clade of *incertae sedis* is placed as sister to SV and includes Treubarinia, *Golenkinia*, *Jenufa*, as well as *Microspora, Parallela* and *Dictyochloris*.

The BAli-Phy analysis (Fig. S3) yielded a topology quite different from the others, though in terms of Sphaeropleales monophyly more consistent with the cp aa analyses. In Fig. S3, Volvocales are weakly monophyletic with the inclusion of *Spermatozopsis* (BPP 0.63). The Volvocales clade without *Spermatozopsis*, which is well supported in most other analyses, only received BPP of 0.36. Treubarinia, again strongly monophyletic, were found sister to Volvocales + *Spermatozopsis*, and Sphaeropleaceae were sister to Volvocales and Treubarinia (though only supported by BPP of 0.18). The grouping of Volvocales, Treubarinia, Sphaeropleaceae, *Jenufa* and *Golenkinia* received a BPP of 0.87. Scenedesminia (supported by 0.84) grouped with Microsporaceae + *Dictyochloris* and their relationship was supported by 0.88 BPP.

Although we could not analyze the order of the genes in the cp genome because most of the newly presented data are not fully assembled genomes, we were able to note the presence/absence of genes (with some missing data, Table S1). A number of cp genome features enforce the deep split of OCC and SV clades, as well as define some of the major orders within these two clades (Fig. 4). All OCC species examined lack a *petA* gene in their cp genomes, and possess a trans-spliced *petD* gene and a trans-spliced *psaC*. Notably, a similar group II intron also appears in the *psaC* of three taxa outside of the OCC clade, *Dictyochloris*, *Elakatothrix*, and *Borodinellopsis*, but is likely cis-spliced (this is uncertain because the data are incomplete). All members of the SV clade lack a *psaM* gene and all but one species (*Carteria cerasiformis*) have a trans-spliced *psaA* gene. Further, all members of Volvocales (with the exception of the *incertae sedis Spermatozopsis*) lack an *infA* gene (in common with members of Chaetophorales in the OCC lineage).

## DISCUSSION

### Chloroplast genome evolution in the Chlorophyceae

The enhanced taxon sampling in our data set was useful in showing several genomic features with peculiar evolutionary histories, despite the fact that most of our new genome data

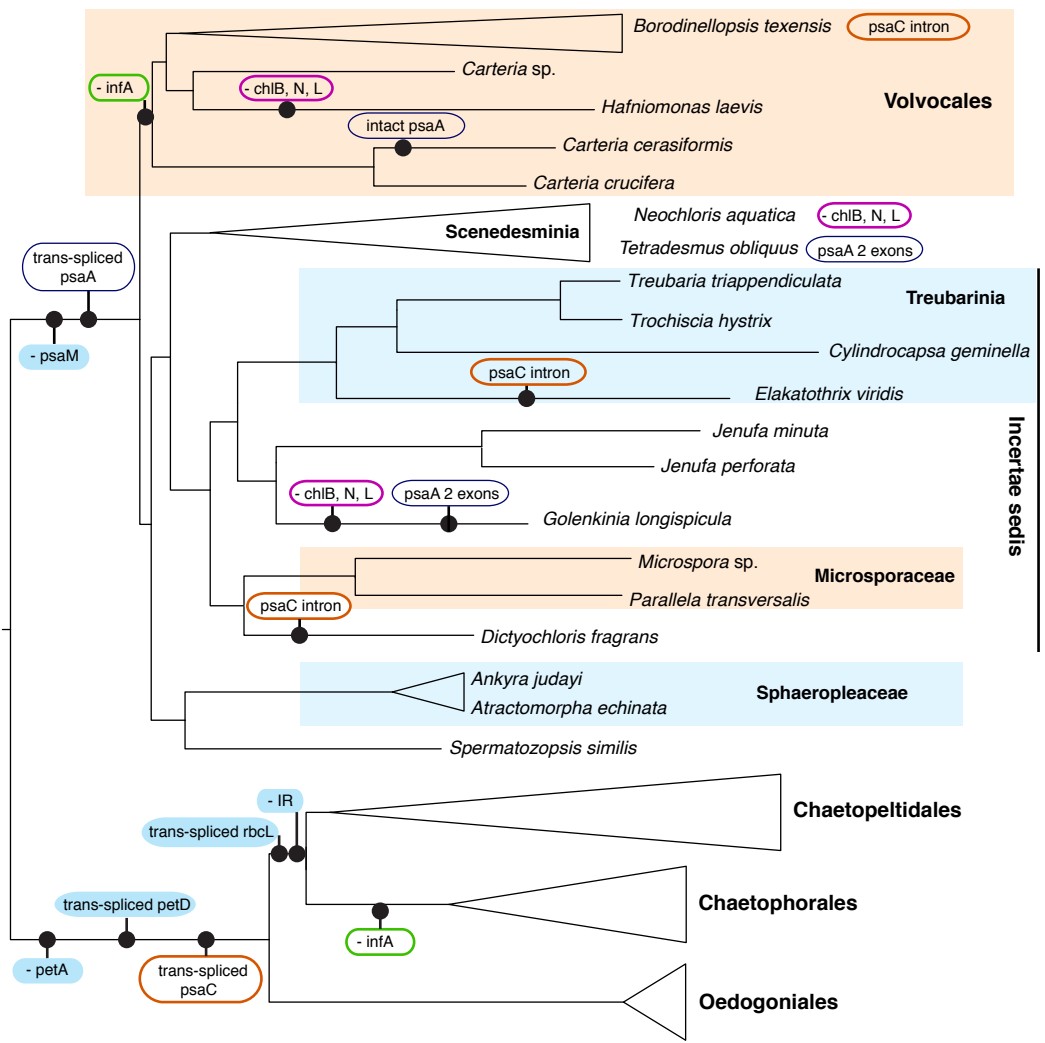

**Figure 4** Schematic representation of the phylogenetic history of focal chlorophycean green algae (as in Fig. 2) showing major gene losses ("-") and modifications ("trans-spliced") in their chloroplast genomes. The nucleotide-based topology was used, but the number of rearrangements would not change if the alternative topology was used. The following features are assumed present in the common ancestor (inverted repeat, *petA*, *infA,* the LIPOR complex *chlB, chlN,* and *chlL, psaM,* and orthodox *petD, psaA, psaC, rbcL*). Unique events, those without reversals or parallel losses, are shown in filled blue ovals. Variously colored unfilled ovals represent modifications of different genes that have evolved at least twice in Chlorophyceae. IR denotes the inverted repeat. Note that the three gene LIPOR cassette (*chlB, chlN, chlL*) is lost from the cp genome from taxa in three major groups.

were not complete. In Fig. 4 (see Table S1 for detailed information), the *psaC* gene appears to have acquired a group II intron multiple times. This intron is inserted at position 25 in the chlorophyceans *Borodinellopsis texensis*, *Jenufa perforata* and *Carteria* sp. and is quite large, spanning over 6.6 kb in *Borodinellopsis*, 2.4 kb in *Carteria*, and 1.5 kb in *Jenufa* despite containing no detectable open reading frames (cutoff 500 bp) or protein domains. A group II intron in the same *psaC* position was also detected in *Dictyochloris* and *Elakatothrix*,

but due to the partial nature of the data it is unclear how large the intron is, or if it is cis-spliced. *PsaC* also has a trans-spliced group II intron at position 25 in all studied OCC taxa –which is one of the genomic-structural synapomorphies for the clade. In all other included Chlorophyceae the *psaC* gene is intact, suggesting that these intron insertions have occurred independently several times. Such independent acquisition of an intron is consistent with the findings of others, e.g., *McManus et al. (2012)*. Additional examples of dynamic intron movement are likely to emerge from chloroplast data, although it is in some cases difficult to determine whether the data represent multiple acquisitions or multiple losses of introns.

Perhaps most curious is the case of *psaA*, which is trans-spliced in the SV clade. The one salient exception is *Carteria cerasiformis*, which has a completely intact (or orthodox) *psaA*. An analysis of the *psaA* gene data did not reveal any indication of horizontal gene transfer unless from a close relative, and examination of the published genome did not reveal introns or other extra-genic DNA that may indicate joining of formerly separated portions. Thus, this apparent re-joining of exons remains unexplained, though perhaps not unique in Chlorophyceae (e.g., the mitochondrial *rns* gene in *Neochloris* in Fučíková et al., 2014). While most SV taxa have three trans-spliced *psaA* exons, the first two exons appear re-joined in *Tetradesmus obliquus* and *Golenkinia longispicula*. A retroprocessing mechanism behind genomic reconnection was recently proposed by *Grewe, Zhu & Mower (2016)*, who discussed an analogous example in a mitochondrial gene of a vascular plant (*Pelargonium*, Geraniaceae). In general, trans-spliced arrangement is considered prohibitive of intron loss, but green algae might provide further exceptions to this rule.

The loss of *infA* is common to Volvocales and to Chaetophorales, providing a further example of convergent gene loss from the plastome. The most striking example, however, is the loss of the light-independent protochlorophyllide oxidoreductase (LIPOR) gene cassette (*chlB*, *chlL* and *chlN*) from *Hafniomonas laevis* and *Neochloris aquatica*, and also possibly from *Golenkinia longispicula*. The incomplete nature of the data for the latter taxon leave some room for doubt, but all other expected genes were recovered and reported from *Golenkinia* by *Lemieux et al. (2015)*. The loss of the LIPOR genes from plastids of diverse algal groups, including red algae and other classes of green algae (*Hunsperger, Randhawa & Cattolico, 2015*) appears correlated with multiple copies of the nuclear-encoded *por* gene (light-dependent protochlorophyllide oxidoreductase). Nuclear genomes of the pertinent Chlorophyceae are unfortunately not available at this time, so we can only speculate whether the LIPOR genes were lost from *Hafniomonas*, *Neochloris*, and *Golenkinia* completely, and if *por* genes occur as single or multiple copies in these algae.

## Increased character and taxon sampling as a phylogenetic remedy

The obvious advantage of whole plastome data is the great number of nucleotide characters available for phylogenetic inference. The obvious drawback is that there are still relatively few green algal taxa with whole cp genome sequences available. The case is reverse for the 18S marker: many taxa have an 18S accession in GenBank, but the entire gene is only ca. 1,800 base pairs long. In addition, not all of the gene's nucleotides are phylogenetically informative, and some cannot be confidently aligned across distantly related taxa. Further,

many 18S GenBank accessions contain partial data, only comprising a few hundred nucleotides. We have attempted a direct comparison of 18S and plastid data using the same taxon set (Fig. 2 and Fig. S3) and found the 18S phylogeny to be quite different from the plastid trees (e.g., Table 2), though often consistent with previous 18S-based studies. The character-rich plastid data set provided strong (often absolute) support for most relationships in our trees—but different analyses yielded different, and well-supported, topologies. Thus, simply adding more characters does not necessarily increase confidence in a phylogeny—much of the informative variation appears restricted to the shallow splits of the tree (Fig. 3), and the high support shown in concatenated trees may raise false confidence in the deep splits.

The problematic relative arrangement of *Pediastrum, Neochloris* and *Chlorotetraedron* (representatives of the families Hydrodictyaceae and Neochloridaceae) was explored in *Fučíková, Lewis & Lewis (2016)* without a satisfying explanation for the conflicting topologies in different analyses. Similarly, here we recovered all three possible groupings of the three taxa (Fig. 2 and Figs. S1–S5). Most recently this issue received extensive attention in *McManus et al. (2018)*, who expanded the taxon sampling in Hydrodictyaceae to 14 species in five genera, but still recovered conflicting relationships between Hydrodictyaceae and Neochloridaceae in different single-gene and concatenated analyses. Expanding the sampling of Neochloridaceae, especially to include the genus *Tetraedron*, is the next logical step in the effort to resolve this systematic problem. However, given the varying signal from individual chloroplast and mitochondrial genes reported in *McManus et al. (2018)*, it is possible that another confounding factor needs to be considered, such as horizontal gene transfer or insufficient model complexity in analyses. For the scope of our study, positioning of the two families is a minor issue, but the results of *McManus et al. (2018)* exemplify that increased taxon sampling may not always be a sufficient solution to all phylogenetic conflict; adding species helped within Hydrodictyaceae but did nothing to resolve the relationships among closely related families. Indeed, our main goal was to add resolution to the chlorophycean tree by increasing taxon sampling especially of deeply diverging *incertae sedis* lineages, but instead of converging on a single topology, we present several viable phylogenetic hypotheses. To our knowledge, we have sampled the known *incertae sedis* thoroughly, but perhaps species that would 'break up' additional long branches in the tree are yet to be discovered, or they may be extinct.

## Unexpected and problematic lineage placements

The filament-formers *Parallela transversalis* and *Microspora* sp. grouped strongly together (Fig. 2, Supplementary Data). Our analyses either placed Microsporaceae along with their apparent coccoid relative *Dictyochloris* in the proximity of Scenedesminia (all aa cp analyses and the 18S analysis), or within the *incertae sedis* clade (all nt cp analyses). Previously, *Parallela* was placed in the proximity of Volvocales by *Novis et al. (2010)* based on data from three plastid genes, but because our study offers a better sampling of deeply diverging chlorophyceans combined with genome-scale sequence data, our results are likely more realistic, even if ambiguous. The family Microsporaceae was listed as member of Sphaeropleales by *Tsarenko (2005)*, but as discussed in *Fučíková, Lewis & Lewis (2014a)*,

verifying this placement will be difficult due to the lack of type material for *Microspora*. Indeed the strain UTEX LB472 used in our study is not a type, and does not even have a species-level identification, so it should be viewed with caution as representative of Microsporaceae.

The respective affiliations of *Spermatozopsis* and *Dictyochloris* are fairly robust based on our analyses, but seem curiously at odds with the current understanding of these taxa based on morphology and ultrastructure. *Spermatozopsis* is traditionally viewed as member of Volvocales because of the clockwise orientation (CW) of its flagellar apparatus (*Melkonian & Preisig, 1984*). Among our analyses, only the BAli-Phy and PhyloBayes nt analyses placed *Spermatozopsis* at the base of Volvocales, separate from Sphaeropleaceae (Figs. S3 and S5), consistently with 18S analyses of *Lewis (1997)*, and *Němcová et al. (2011)*. Curiously, *Marin*'s (*2012*) nu rDNA analysis, using a closely related species *Spermatozopsis exsultans*, recovered this genus sister to Sphaeropleaceae, like our cp analyses did, but his cp rDNA analysis and the combined nu and cp rDNA analysis recovered the genus at the base of Volvocales with representatives of *Carteria* (also see simplified trees in Fig. 1) The presence of the gene *infA* in the cp genome of *Spermatozopsis* contrasts to its absence in all other volvocalean taxa examined, including the deeply diverging *Hafniomonas* and *Carteria*. *Dictyochloris*, in turn, invites placement in Sphaeropleales based on its flagellar ultrastructure nearly identical to that of *Bracteacoccus*, as reported by *Watanabe & Floyd (1992)*. However, the unusual parallel configuration of the basal bodies in both taxa, similar to the volvocalean *Heterochlamydomonas*, led to further investigations by *Shoup & Lewis (2003)* who, again using 18S, placed *Dictyochloris* squarely in Sphaeropleales but unrelated to *Bracteacoccus,* and interestingly also showed Treubarinia as sister to Sphaeropleaceae. The phylogenetic tree of *Němcová et al. (2011)*, however, places *Dictyochloris* sister to *Microspora*, consistently with our Fig. 2.

Another "straggler" taxon subtended by a long, weakly placed phylogenetic branch has previously been the coccoid genus *Mychonastes* (e.g., *Krienitz et al., 2011*). Whereas *Fučíková, Lewis & Lewis (2014a)* showed the genus at the base of Sphaeropleales, in none of our scenarios was *Mychonastes* the deepest diverging branch of the order. The lineage appears as a long-branched member of Scenedesminia without a particularly close alliance to other families.

## Sphaeropleales—greatly broadened or greatly narrowed?

Though the above described phylogenetic results are intriguing, our primary focus is the inconsistently recovered monophyly of Sphaeropleales, which has far-reaching taxonomic implications (Fig. 1). As shown in Table 2, the clade designated here as Scenedesminia is phylogenetically quite robust. In fact, aside from Treubarinia it is our only focal group that appears unaffected by analysis type, and is supported by 18S data as well. Sphaeropleaceae, while morphologically quite different from Scenedesminia, share similarities in flagellar apparatus ultrastructure with them (*Hoffman, 1984*; *Watanabe, Floyd & Wilcox, 1988*; *Wilcox & Floyd, 1988*). Yet, molecular support for a monophyletic Sphaeropleales incl. Scenedesminia has been weak in previous studies, especially where 18S was the sole marker used (e.g., *Shoup & Lewis, 2003*). The support improved with the use of whole plastome

protein coding data (*Fučíková, Lewis & Lewis, 2016*), even though not all genes agreed on the order's monophyly. Despite building on the study of *Lemieux et al. (2015)* by including additional *incertae sedis* Chlorophyceae, our study does not bring the desired order to Sphaeropleales. Instead, Sphaeropleaceae, Treubarinia, and various other deeply diverging chlorophycean taxa vary in their placement across analyses (Table 2). Most of our cp nt phylogenies agree on Sphaeropleales *s. l.*, which would include Microsporaceae, Dictyochloridaceae and Sphaeropleaceae as previously noted (*Tsarenko, 2005*) along with Scenedesminia. The hypothetical broadened order further includes Treubarinia (previously similarly placed by *Shoup & Lewis (2003)* and corroborated by *Lemieux et al., 2015*), *Jenufa* and *Golenkinia*, and surprisingly also *Spermatozopsis similis*. The cp nt trees disagree on the branching order within Sphaeropleales *s. l.*, but the Bayesian amino acid phylogeny (Fig. 2B), both PhyloBayes trees (Figs. S4, S5), and the BALi-Phy 18S tree (Fig. S3) contradict this arrangement entirely, rendering Scenedesminia order-less and instead grouping Sphaeropleaceae at or near the base of Volvocales.

The PhyloBayes results hint at the possibility of site-specific evolutionary patterns in the nucleotide data, which the CAT model specifically tries to address (*Lartillot & Philippe, 2004*). SVDquartets, in turn, aims to account for incomplete lineage sorting and the resulting conflict among signals from different genes (*Chifman & Kubatko, 2014*; *Chifman & Kubatko, 2015*). Even though plastid genes are often assumed to evolve as a single locus, among-gene conflict has been demonstrated in previous studies (e.g., *Fučíková, Lewis & Lewis, 2016*; *McManus et al., 2018*), suggesting the possibility of more complex inheritance patterns. Lateral gene transfer is a possible explanation for among-gene conflict, but our data are at present insufficient to support or reject this possibility. Our SVDquartets tree (Fig. S2) supports the most "traditional" view of distinct Volvocales and Sphaeropleales, with Sphaeropleaceae and Scenedesminia grouping together, though also including *Spermatozopsis* as sister to Scenedesminia. However, neither of the two orders is well supported—both receive less than 50% BS—and therefore this result cannot be viewed as definitive. The SVDquartets analysis also strongly groups all *incertae sedis* taxa: Treubarinia, Microsporaceae, *Dictyochloris*, *Golenkinia* and *Jenufa* receive 97% BS as a clade. This grouping appears in several other analyses but mostly embedded within Sphaeropleales *s.l.* (Table 2, Fig. 2A, Figs. S1–S6).

It is possible that other factors contribute to the observed conflicts in phylogenetic signal. Saturation of 3rd positions is often blamed for phylogenetic issues, for example. However, in our case, this problem is unlikely to apply: saturation implies that the rate of evolution is so high that no historical signal is present. Saturated sites represent noise and thus are expected to reduce support overall. They should not strongly support any particular clade, nor should they conflict strongly with others sites that have a strong signal. We show this in our supplementary analyses: 3rd positions do not have much signal at the critical deep nodes (manifested by low BPP and polytomies), so they are unlikely to cause conflict. On the other hand, 3rd positions are very informative at shallow divergences, and thus likely contribute to good resolution in many parts of the concatenated tree.

One concern is that saturated 3rd position sites are not as affected by stabilizing selection and thus might have a different base composition than 1st or 2nd position sites. This is

effectively handled by partitioning (as implemented in our analyses), which allows 3rd position sites to have their own nucleotide frequencies. Moreover, assessing saturation is problematic in itself, as "saturation plots" comparing model-corrected vs. uncorrected distances rely on pairwise comparisons. These comparisons are misleading because the largest distances span the entire tree using data from only two taxa. Bayesian and maximum-likelihood methods for estimating phylogeny instead estimate much shorter individual edges of the tree using all the data. Genes or data subsets (e.g., 3rd codon positions) that appear to be saturated using pairwise saturation plots may be evolving at a rate ideal for maximum-likelihood or Bayesian approaches. Evaluation of saturation is an area much in need of better methods. In summary, given the disparity of results among different analyses, the class Chlorophyceae clearly has a history of complex evolutionary processes that currently cannot all be accommodated in a single analysis.

It is entirely possible that with the addition of even more data (both in terms of taxa and genes), Sphaeropleales will be shown as a robust order. At present, however, it is at least equally possible that Sphaeropleales contains only one family, the Sphaeropleaceae. This would mean that Scenedesminia, and perhaps other lineages, be considered distinct orders as well. In our view, given the phylogenetic uncertainty, such drastic taxonomic changes do not seem justified. The problems with solidly anchoring the order-defining family Sphaeropleaceae in the chlorophyte phylogeny could lead to years of taxonomic changes and redefinitions in the future. Therefore, we adopt a practical non-Linnaean solution with well-supported clades as taxonomic currency instead (*Nakada, Misawa & Nozaki, 2008*). We consider Scenedesminia and Treubarinia useful names for the two well-supported groups. In addition, within Scenedesminia the clade of mostly colony-forming families (Hydrodictyaceae, Neochloridaceae, Scenedesmaceae and Selenastraceae, plus the coccoid Rotundellaceae) was well supported in nearly all our analyses. In the future, this 'colonial clade' might warrant a formal recognition, but here we merely note it as an interesting hint at a pattern in morphological evolution. In general, rather than make formal taxonomic changes in the Linnaean framework, we prefer the approach to name clades, as has been the practice in phycology for a long time, e.g., in Trebouxiophyceae, where order- and family-level relationships have long been unclear and a clade system has been in use (e.g., *Neustupa et al., 2013*; *Fučíková, Lewis & Lewis, 2014b*).

## CONCLUSIONS

Whole plastome analyses of a total of 68 taxa across the green algal class Chlorophyceae yielded well-supported topologies, which however differed depending on the type of analysis. In particular, the Bayesian analysis of the amino acid data yielded a non-monophyletic order Sphaeropleales, rendering most of its families in potential taxonomic limbo. Two mutually exclusive taxonomic conclusions could be drawn based on our data, each comprising an ordinal-level revision. The first is broadening Sphaeropleales to include *incertae sedis* chlorophyceans, as most nucleotide analyses suggest. The other is narrowing Sphaeropleales to only include Sphaeropleaceae, and erecting at least one, but likely multiple new orders in Chlorophyceae to accommodate the remaining lineages.

For practical purposes, we propose the Sphaeropleales outside of Sphaeropleaceae to be treated as a clade called Scenedesminia. Aside from the well-supported clade Treubarinia, the *incertae sedis* taxa do not always form robust groupings and are best taxonomically handled as individual genera without higher affiliation.

## ACKNOWLEDGEMENTS

Analyses were carried out at the Computational Biology Core Facility of the University of Connecticut.

### Funding

Data collection and postdoctoral support for Karolina Fučíková were funded by the NSF grants DEB-1036448 and DEB-1354146 awarded to Louise A. Lewis and Paul O. Lewis. The funders had no role in study design, data collection and analysis, decision to publish, or preparation of the manuscript.

### Grant Disclosures

The following grant information was disclosed by the authors:
NSF grants: DEB-1036448, DEB-1354146.

### Competing Interests

The authors declare there are no competing interests.

### Author Contributions

- Karolina Fučíková conceived and designed the experiments, performed the experiments, analyzed the data, prepared figures and/or tables, authored or reviewed drafts of the paper, approved the final draft.
- Paul O. Lewis and Louise A. Lewis conceived and designed the experiments, analyzed the data, contributed reagents/materials/analysis tools, prepared figures and/or tables, authored or reviewed drafts of the paper, approved the final draft.
- Suman Neupane analyzed the data, prepared figures and/or tables, approved the final draft.
- Kenneth G. Karol analyzed the data, contributed reagents/materials/analysis tools, approved the final draft.

### DNA Deposition

The following information was supplied regarding the deposition of DNA sequences:

Complete and incomplete genomes of the newly sequenced algae are deposited via GenBank under accession numbers MG778120-27, MG778173, MG778174-84, MG778185, MG778230-34, MG778446-90, MG778491-99, MG778351-407, MG778235-96, MG778408-09, MG786420, MG778500, MG778186-229, MG778297-350, MG778501-32, KT693210-2, MG778128-72, MG778410-45, and MG778533.

## Data Availability

Raw data are available in the Supplemental Files.

## Supplemental Information

Supplemental information for this article can be found online at http://dx.doi.org/10.7717/peerj.6899#supplemental-information.

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
