# Peer review of "Order, please! Uncertainty in the ordinal-level classification of Chlorophyceae"

_PeerJ, doi:10.7717/peerj.6899_

## Round 0.1 · original submission · Minor Revisions

The paper received contradictory remarks from three reviewers. It needs revision. However please particularly take into account the comments from the second reviewer.

Reviewer 1 ·

Basic reporting

The authors sequenced the chloroplast genome of 18 incertae sedis or deeply branching Chlorophyceae species and used a chloroplast phylogenomic approach to solve the phylogenetic relationships between Chlorophyeceae orders. Despite that three different and complementary state-of-the-art analyses were performed, the results did not converge on a single topology, but two mutually exclusive and equally likely phylogenetic scenarios were recovered. Both scenarios are thoroughly discussed, in the light as well of chloroplast genomes organization and evolution.

The manuscript is well written in clear, unambiguous and professional English and the structure conforms to PeerJ standards. Objectives, methods and results are clearly presented and the discussion is well reasoned. The figures are of high quality, appropriately described and labelled and they support and complement the text. The raw data and the custom script are available and accessible.

Experimental design

This manuscript presents original primary research within the scope of PeerJ journal. The research question is clearly defined and circumscribed and aimed at closing an existing knowledge gap. The dataset generated is meaningful and comprises of a well-balanced taxon sampling. The authors methodology is rigorous and state-of-the-art approaches were used in conformity with the prevailing technical standards in the field. The description of materials and methods are exhaustive and, together with the raw data and the scripts made available, ensure the replicability of this study.

Validity of the findings

The dataset is robust and the taxon sampling is exhaustive. Some among the best softwares available have been used for a rigorous and comprehensive data analysis (ML, Bayes, coalescence-based). However, the analyses did not converge toward a consensus topology. This is often the case in many phylogenetic studies, where similar datasets result in contrasting and alternative conclusions. Nevertheless, the authors discuss properly the likelihood of the two scenarios, and they link them with chloroplast genome organization, both based on the published literature and on the novel results presented in the manuscript. The conclusions are well stated and they link with the original question investigated.

Additional comments

I found the manuscript well written, well-structure and the methods are robust and comprehensive. I only have minor comments.

1. In my version of the manuscripts, caption of Figure 2, Figure 3, Figure 4, Table 1 and Table 2 report “Microsoft Word – 105” at the beginning of each paragraph. I guess is a typographic misprint.

2. I am aware that Chlorophyceae chloroplast genomes often harbor a substantial fraction of repetitive regions, which impair the performance of assemblers. I would like to suggest you, for your future studies, to use a better suited software for assembling your reads (e.g.: SPAdes, Bankevich, 2012), instead of Geneious. You may get less contigs (or even one contig) for each chloroplast genome.
You may like as well to identify additional chloroplast non-coding contigs, if present, by binning your assembly according to the k-mer composition (e.g.: with MyCC, Lin, 2016). Using this strategy, you will probably recover as well the mitochondrial genome, depending on the sequencing depth of the libraries.

Reviewer 2 ·

Basic reporting

The authors investigated in this study the relationship of the green algal class Chlorophyceae at ordinal level using concatenated data set of 58 protein-coding genes. They used these plastid-coding genes to test the robustness of five 'traditional' orders (Sphaeropleales, Volvocales, Oedogoniales, Chaetophorales, Chaetopeltidales). The analyses for these tests were conducted using amino acid and nucleotide sequences.

The manuscript is mainly well-written, but starting with the title it contains sometimes sloppy terms. The background literature is not sufficient cited. The used traditional orders are not introduced, which were originally based solely on morphological and ultrastructural features. Molecular phylogenetic analyses revised these orders, but they were never formally described and alternative grouping in orders equally valid. Most phylogenetic studies using nuclear genes such as 18S and ITS are not cited (Nakada et al. 2008, Pröschold et al. 2001). The naming of the different lineages in the figures is mixed and confusing. Sometimes the authors used family names, sometimes names based on the PhyloCode and sometimes simply clades named after a representative, all not properly defined (for example, in Fig. 1, no introduction is given about the origin of these trees). The photograph shown in Fig. 1 are of low quality and should be replaced.

The main topic of this study is the ordinal structure of the Chlorophyceae. The authors demonstrated conflicting scenarios in their phylogenetic analyses using different methods and different alignments (amino acid and nucleotide) and discussed these results in the manuscript. However, no possible explanation for these conflicts is given.

Experimental design

The new chloroplast genomes of 18 relevant taxa close some gaps in phylogeny. All raw data are available as supplement. All analyses are of high standard, but several additional analyses are needed. For example, the authors found discrepancies between the phylogenies using amino acid and nucleotide alignment. Are these caused by the saturation of the third codon bases? This should be tested. It should also be mentioned which gene or taxon causes conflicts. Alternative tree topologies should be tested using approximately unbiased tests, which are implemented for example in CONSEL. In addition, it should also be demonstrated if adding of 18S data would increase the support of the topology.
A minor point: Bayesian support below 0.95 mean no support. Therefore, values below this level should be excluded in the figures and replaced by a '-'.

Validity of the findings

As mentioned above, the authors demonstrated that there is a conflict in the robustness of the orders of the Chlorophyceae. The validity of these findings is not discussed in context of other nuclear genes (18S, ITS, 28S) and non-molecular data (morphology, ultrastructure, physiology). It should also be explained why single-gene topologies are different of those of concatenated data sets. It is possible that some plastid-coding genes need to adapt faster than others or be exchanged by horizontal gene transfer as described for prokaryotes? Some explanations and conclusions should be given for these conflicts in the tree topologies.

Additional comments

By addressing the points highlighted above, the manuscript can be accepted for publication, however, present form I recommend 'Major Revisions'.

·

Basic reporting

This excellent paper analyses phylogenetic relationships within the Chlorophyceae based on a taxon-rich dataset of chloroplast genomes. The dataset includes 18 newly sequenced species, and fills several important sampling gaps, in particular in the Sphaeropleales and Volvocales. The analyses support several previously inferred phylogenetic relationships within the Chlorophyceae, but a number of relationships remain unresolved, including the positions of the Treubarinia and Sphaeropleaceae.

The text is clearly written and the figures are of high quality. The introduction provides a good overview of the background and relevant literature is cited. Methods and Results are clear. Some parts of the Discussion are repeated from the Results, and could be moved to the Results section.

Experimental design

The research is novel, and research questions are well defined and relevant. It is clearly stated how the research fills several important sampling gaps for the phylogeny of Chlorophyceae, notably in the Sphaeropleales and Volvocales.

The phylogenetic analyses are of a high technical standard, and the methods are clearly described.

Details on algal culturing conditions, DNA extraction, library preparation, sequencing and annotation are not provided, but can be found in Fučíková et al. (2016), as indicated in the text. For clarity, or for those that have no access to the publication these methods could be provided in supplementary material.

One of the aims of the paper was to assess the presence/absence of genes to study the evolution of gene loss. However, it is difficult to prove the absence of a gene in incompletely assembled genomes, as the authors write in the manuscript. However, additional analyses, such as similarity searches of a gene of interest against all contigs of an assembly, could confirm or provide additional evidence for the absence of a gene.

The evolution of genomic features is based on a single tree topology. However, the phylogenetic analyses present several alternative phylogenetic hypotheses. I think the choice of topology for the analysis of genomic features should be explained more clearly.

The rationale for the SVDquartets is not entirely clear from the text. In the Methods section it is written that SVDquartets allows for site-specific rate variation in tree inference, while in the discussion it is written that SVDquartets aims to account for incomplete lineage sorting and the resulting conflict among signals from different genes. This should be clarified. Also, how relevant is incomplete lineage sorting in phylogenetic analyses inferred from plastid genome data, where all genes are (normally) linked? The authors write “Even though plastid genes are often assumed to evolve as a single locus, among-gene conflict has been demonstrated in previous studies (e.g., Fučíková, Lewis & Lewis, 2016; McManus et al., 2018). However, in these cited papers, I didn't find compelling evidence for unlinked loci in plastid genomes, and among-gene conflict does not necessarily imply that these genes are unlinked. I think think this should be better explained.

Validity of the findings

The phylogenetic data are robust, and the conclusions are well stated, and linked to the original research question.

Additional comments

Minor comments.
Abstract: “the only class whose monophyly remains uncontested as gene and taxon sampling improves.” The Chlorophyta includes several other (smaller) classes (Pedinophyceae, Chlorodendrophyceae, Mamiellophyceae, …) that are monophyletic. Perhaps specify 'core Chlorophyta'.

Fig. 3. The graphical legend (bottom left) doesn’t show the symbol for medium internode certainty.

Line 153. Sequences of 58 protein-coding chloroplast genes were aligned using the translation-aided algorithm in Geneious v.10. I suppose this is based on a given amino acid cost matrix. Please indicate the parameters used.

---

## Round 0.2 · Major Revisions

Please update the manuscript using text from your answer to the reviewer. The paper needs some revision, including the figures. Please keep in mind wide readers audience of the journal, naming of the lineages should be is easier to follow for a non-phycologist. The remarks are not so critical; so I believe you can re-submit the manuscript soon.

Reviewer 2 ·

Basic reporting

The author made some changes in the ms and responded to my comments. However,
the naming of the lineages is still not changed and difficult to follow for a non-phycologist. Therefore, I strongly recommend to define the lineages by simply naming them according to representatives. As many publication have demonstrated, most classical orders are not monophyletic. One option to present a 18S tree only and name here the clades and orders. This tree could then replace Fig. 1.

Experimental design

no comment

Validity of the findings

All analyses are of high standard, but several additional analyses are needed. For example, the authors found discrepancies between the phylogenies using amino acid and nucleotide alignment. Are these caused by the saturation of the third codon bases? This should be tested.
Response by the authors: Saturation implies that the rate of evolution is so high that no historical signal is present. Saturated sites represent noise and thus are expected to reduce support overall. They should not strongly support any particular clade, nor should they conflict strongly with sites that have strong signal. One concern is that saturated 3rd position sites are not as affected by stabilizing selection and thus might have a different base composition than 1st or 2nd position sites. This is effectively handled by partitioning, which allows 3rd position sites to have their own nucleotide frequencies. We have explored the question of information content in 3rd vs 1st,2nd sites, but prefer to not complicate this paper by including the results. Briefly, we conducted analyses using Phycas that used a flat prior on resolution class, where resolution class is the number of internal nodes (resolution class 1 is the star tree, while the highest resolution class consists of all fully-resolved trees). If site have no information about tree topology, the posterior should also be flat across resolution classes (highest possible entropy). If sites have lots of information, all of the posterior probability is concentrated in the fully-resolved class (lowest possible entropy). This analyses shows that our 3rd position sites have more information than 2nd position sites for all genes, and, in several genes, 3rd position site have more information than even 1st position sites. Thus, the 3rd positions are not saturated, and, even if they were, the codon-level partitioning should have accommodated any deviation in nucleotide frequencies relative to 1st and 2nd position sites.

Comment: This response should be in the manuscript because that could be one reason for the discrepancies between nt and aa tree. If the 3rd position is more informative than the first two, what would be happened if you analyze these positions only. I personally do not believe that this is true. For example, when I checked randomly rbcL or psaA, most 3rd codon positions were saturated.

It should also be mentioned which gene or taxon causes conflicts.
Response by the authors: We address the ‘straggler’ taxa in our Discussion. The gene issue is visualized in the Treespace supplements. Fig. S6, and other supplementary graphs in the Treespace subfolder (esp. for groupings 1 and 4) show that the concatenation yields a very different tree from all single gene trees – that is the main problem, but not entirely unexpected. But among single gene trees there doesn’t seem to be one gene that would be completely apart and at odds with the rest.

Comment: To address this point, the authors also should analyze their data sets by non-tree based methods (i.e. TIGER, Cummins & McInerney 2011: Syst. Biol. 60: 833-844). This would discover if fast-evolving genes cause trouble by the best tree finding.

Additional comments

I still miss some explanations for the discrepancies demonstrated in the ms. I would like to see some explanations for different tree topologies. Is that caused by different evolutionary rates of genes or by biases in the analyzing methods?

·

Basic reporting

This is a solid phylogenetic study. The manuscript is clearly written and nicely illustrated. In my previous review, I only had minor comments, all of which have been satisfactory dealt with. The sequence data on GenBank are not yet available, but I trust that these will be released after publication.

Experimental design

no comment

Validity of the findings

no comment

Additional comments

no comment

---

## Round 0.3 · accepted · Accept

Now all the reviewers are satisfied, no more comments.

# Reviewer 2 ·

Basic reporting

The author made some changes in the ms and responded to my comments. However,
the naming issue of my last review still remained, but I leave it now by the editor to decide if the changes made by the authors are enough for publication.

Experimental design

no comment

Validity of the findings

The authors made substantial changes and added explanations.

Additional comments

There are still issues about the naming but I leave it with editor to decide.

·

Basic reporting

In my review of the first version of the manuscript, I only had minor comments, all of which have been satisfactory dealt with.

Experimental design

no comment

Validity of the findings

no comment

Additional comments

no comment